

# Adaptive adjacent context negotiation network for object detection in remote sensing imagery

Yan Dong[1,2], Yundong Liu[2], Yuhua Cheng[1], Guangshuai Gao[2], Kai Chen[1] and Chunlei Li[2]

[1] School of Automation Engineering, University of Electronic Science and Technology of China, Chengdu, China
[2] School of Electronic and Information Engineering, Zhongyuan University of Technology, ZhengZhou, China

## ABSTRACT

Accurate localization of objects of interest in remote sensing images (RSIs) is of great significance for object identification, resource management, decision-making and disaster relief response. However, many difficulties, like complex backgrounds, dense target quantities, large-scale variations, and small-scale objects, which make the detection accuracy unsatisfactory. To improve the detection accuracy, we propose an Adaptive Adjacent Context Negotiation Network ($A^2$CN-Net). Firstly, the composite fast Fourier convolution (CFFC) module is given to reduce the information loss of small objects, which is inserted into the backbone network to obtain spectral global context information. Then, the Global Context Information Enhancement (GCIE) module is given to capture and aggregate global spatial features, which is beneficial for locating objects of different scales. Furthermore, to alleviate the aliasing effect caused by the fusion of adjacent feature layers, a novel Adaptive Adjacent Context Negotiation network ($A^2$CN) is given to adaptive integration of multi-level features, which consists of local and adjacent branches, with the local branch adaptively highlighting feature information and the adjacent branch introducing global information at the adjacent level to enhance feature representation. In the meantime, considering the variability in the focus of feature layers in different dimensions, learnable weights are applied to the local and adjacent branches for adaptive feature fusion. Finally, extensive experiments are performed in several available public datasets, including DIOR and DOTA-v1.0. Experimental studies show that $A^2$CN-Net can significantly boost detection performance, with mAP increasing to 74.2% and 79.2%, respectively.

# INTRODUCTION

Benefiting from developments in fields like aviation and computers, the quantity and quality of remote sensing satellite imagery have also improved dramatically. Localization and identification of objects of interest in remote sensing images (RSIs) are essential for object identification, resource management, decision-making, and disaster relief response. It plays a catalytic role in military reconnaissance (*Xie et al., 2024*), ecological protection (*Rodofili,*

Corresponding author
Yan Dong, dy@zut.edu.cn

*Lecours & LaRue, 2022*), unmanned vehicles (*Lyu et al., 2022*), urban planning (*Shen et al., 2023*) to name a few. Although many researchers have proposed many algorithms for remote sensing object detection (RSOD), this task still needs to overcome many difficulties, mainly due to the complex backgrounds, dense target quantities, large-scale variations, and small-scale objects. There is still a lot of work to be explored on how to sift through the large amount of information to find useful information.

Unlike natural images (NIs), RSIs are obtained from an aircraft looking down on the ground. Due to the influence of Earth's gravity, objects in natural images generally have specific *a priori* knowledge. For example, cars are parked on the ground with their wheels facing downward, and trees and pedestrians are generally oriented vertically. If this prior knowledge can be properly utilized then the detection performance of the algorithm can be improved. Compared to NIs, RSIs can only yield a top view of the object as they are imaged from a top-down perspective. In addition, RSIs contain much richer information since they consist of many objects of different sizes and shapes on the ground, and the object sizes are usually small. These differences prevent many algorithms proposed for natural images from being well applied directly to RSIs.

Currently, deep learning-based RSOD algorithms mainly draw on the solution ideas of object detection algorithms in generalized scenarios. The mainstream algorithms can be sketchily classified into two types: two-stage methods and one-stage methods. The former accomplishes the detection task in two steps. Firstly, the object regions of interest are extracted, and then these regions are located and classified. For example, *Girshick et al. (2014)* proposed R-CNN in 2014, followed by a family of R-CNN algorithms (*Girshick, 2015*; *Ren et al., 2015*), these algorithms perform well in terms of accuracy but have high computational complexity, which is hardly applicable to devices with limited resources and high real-time requirements. One-stage algorithms directly classify and localize each pixel or region in the image, thus making the detection problem much simpler. One-stage methods have lower computational complexity but may have a slight loss in accuracy. Therefore, one-stage detection algorithms have not only received attention from researchers but have also been widely applied in practical engineering projects Examples include SSD (*Liu et al., 2016*), YOLOv3 (*Redmon & Farhadi, 2018*), EfficientDet (*Tan, Pang & Le, 2020*), YOLOv4 (*Bochkovskiy, Wang & Liao, 2020*), YOLOv6 (*Li et al., 2022*), YOLOSA (*Li & Huang, 2023*), and YOLOv7 (*Wang, Bochkovskiy & Liao, 2023*), *etc.* Although these algorithms perform well on generalized natural scenes, they are poor performers in direct migration applications on RSIs, especially for small dense objects and multi-scale objects.

Considering RSIs' various characteristics, many algorithms for RSOD tasks have been proposed. For instance, *Shi et al. (2021)* combined deconvolution with position attention to capture the external and internal feature information of the aircraft during feature generation, respectively. *Zhang et al. (2022)* enhanced the feature representation of the detector through internal knowledge (*e.g.*, feature similarity, spatial location) and external knowledge (*e.g.*, co-occurrence, intrinsic properties). *Huang et al. (2023)* adopted an offset subnet to predict the offset position map on the query feature map, then sampled the most relevant features from the input feature map based on the offset positions, and finally used the sampled features to compute the self-attentive key and value maps and reconstruct the

feature map. *GU et al. (2023)* stacked deformable convolution and adaptive spatial attention in tandem through dense connections alternately and constructed a dense context-aware module capable of modeling local geometric features to achieve the relationship modeling between target and scene content based on fully utilizing the semantic and positional information of different layers. *Huang, Tian & Li (2023)* exploited global context-aware object representations and fine-grained boundary structures to complement feature information.

However, the performance of these methods in complex scenes is not satisfactory, especially for small objects in RSIs. Recent studies have demonstrated that rich contextual information plays a key role in detecting visually poor objects, such as small or occluded objects. A common approach is to utilize the attention mechanism to target attention to the regions where small objects are located by assigning different weights to different regions in the image. In addition, multi-scale feature extractors can be introduced to capture object information at different scales and combine them with contextual information for comprehensive analysis. For instance, *Cheng et al. (2020)* proposed a cross-scale feature fusion detector for feature fusion and feature enhancement at each feature scale. *Liu et al. (2021c)* proposed an adaptive FPN to capture more discriminative features, which combine multi-scale features across different channels and spatial locations. *Dong et al. (2022)* used feature pyramid networks and dilated convolutions to fuse contextual information in multi-scale features. *Zhang et al. (2019)* and *Wang et al. (2022)* also draw on the strategy of contextual information modeling to handle sophisticated scene issues and enhance detection performance. *Huo et al. (2023)* replaced the intersection over union with the normalized Wasserstein distance, which effectively mitigates the issue that extended metrics based on intersection over union are very sensitive to small object positional deviations. Although most of these algorithms improve the detection accuracy to a certain extent, they often use upsampling and element-wise summation for adjacent feature layers to merge feature maps of different scales. This makes feature confusion occur in feature fusion between adjacent feature maps and also introduces interference information in feature maps.

To this end, we propose an efficient detector based on an adaptive adjacent context negotiation network ($A^2$CN-Net), which contains a series of components to boost detection performance. Initially, to extract richer features of small objects, a composite fast Fourier convolution module is given based on fast Fourier transform and inserted into multiple stages of the backbone network to capture spectral contextual information. Then, a global context information enhancement module is given to capture and aggregate global spatial features, which is beneficial for locating targets of different sizes. Furthermore, to alleviate the aliasing effect caused by the fusion of adjacent feature layers, a novel adaptive adjacent context negotiation module is given to the adaptive integration of multi-level features, which consists of local and adjacent branches, with the local branch adaptively highlighting feature information and the adjacent branch introducing global information at the adjacent level to enhance feature representation. In the meantime, considering the variability in the focus of feature layers in different dimensions, learnable weights are applied to the

local and adjacent branches for adaptive feature fusion. From our experimental results, $A^2$CN-Net can significantly boost detection performance for the RSOD task.

The key contributions of this work are summarized as follows.

- We propose an adaptive adjacent context negotiation network ($A^2$CN-Net), which can significantly boost detection performance on RSOD datasets.
- To reduce the information loss of small objects in deep networks, the composite fast Fourier convolution module is given, which is inserted into the backbone network to obtain spectral global information.
- To strengthen feature information representation, a global context information enhancement module is given, which captures and aggregates global and local spatial feature information.
- To alleviate the aliasing effect caused by the fusion of adjacent feature layers, an adaptive adjacent context negotiation network is given, which adaptively combines contextual feature information.

## RELATED WORK

### Remote sensing object detection

Early RSOD models typically relied on hand-crafted features or pre-existing geographic information as a way to filter candidate areas. For instance, *Liu et al. (2012)* used template matching to find out the areas where the aircraft might exist and then utilized principal component analysis and kernel density function to identify the potential regions. *Yao et al. (2015)* used the Hough transform to judge whether there is a potential airport and then used the method based on salient region extraction to extract SIFT features from the candidate region. *Gu, Lv & Hao (2017)* used Markov Random Fields to model the positional relationship between objects in space to classify objects. However, these methods used hand-designed features or analyzed them based on previous geographic data to find areas relevant to the target. Such methods are limited by the limitations of hand-crafted features and the availability of prior geographic information. To overcome these limitations, in recent years, as shown in Table 1, many research efforts have begun to explore deep learning-based RSOD models according to the characteristics of RSIs.

For multiscale characteristics of RSIs, *Li et al. (2019)* aggregated feature information of global spatial locations on multiple scales by introducing self-attention. *Zhang, Lu & Zhang (2019)* design spatial and scale-aware attention modules to direct the network's attention to more information-rich regions and features, as well as more appropriate feature scales. *Xu et al. (2021)* proposed a context-based feature alignment network, which can effectively correct the misalignment between convolution kernel sampling points and objects, thereby improving feature consistency. *Zhang et al. (2023)* employed a global-local feature enhancement module to address the scale variation issue by capturing local features with multiple receptive domains through pooling operations and obtaining global features through non-local blocks. This module effectively captures local details and preserves global contextual information, which contributes to more accurate and robust classification.

**Table 1  Some deep learning-based algorithms for remote sensing object detection.**

| Challenges | Model | Dataset | Detection accuracy (mAP%) |
|---|---|---|---|
| Large scale variations | *Li et al. (2019)* | DOTA–HBB (*Xia et al., 2018*) | 75.38 |
| | *Zhang, Lu & Zhang (2019)* | DOTA–HBB | 69.9 |
| | *Xu et al. (2021)* | DIOR (*Li et al., 2020*) | 71.1 |
| | *Zhang et al. (2023)* | DIOR | 73.8 |
| Small objects | *Wang et al. (2019)* | DOTA–HBB | 72.43 |
| | *Lingyun, Popov & Ge (2022)* | DIOR | 73.8 |
| | *Chen et al. (2023)* | DOTA–HBB | 63.02 |
| | *Yang & Wang (2024)* | DIOR | 73.9 |
| Arbitrary directions | *Cheng et al. (2022a)* | DIOR-R | 64.41 |
| | *Cheng et al. (2022b)* | DIOR-R | 65.1 |
| | *Yao et al. (2023)* | DIOR-R | 64.2 |
| | *Cheng et al. (2023)* | FAIR1M-1.0 (*Sun et al., 2022*) | 40.7 |

To increase sensitivity to small objects in RSIs, *Wang et al. (2019)* proposed a re-weighted loss function to pay more attention to small. *Lingyun, Popov & Ge (2022)* introduced frequency domain convolution to extract richer small object features by sensing spectral context information, thus enabling more accurate and detailed classification. *Chen et al. (2023)* proposed a Multiple-in-Single-out feature fusion structure to enhance local information interaction and used adaptive Intersection Over Uni-Tiny loss to enhance the positioning accuracy of small objects. *Yang & Wang (2024)* devised super-resolution networks incorporating soft thresholding techniques to enhance small target features, thereby elevating the resolution of the feature map while minimizing redundancy.

For the object rotation variations, *Cheng et al. (2022a)* produced coarse-oriented boxes by a coarse location module in an anchor-free manner and then refined them into high-quality oriented proposals. *Cheng et al. (2022b)* designed the localization-guided detection head to mitigate the feature mismatch between classification and localization to improve the accuracy and robustness of the model. *Yao et al. (2023)* proposed a simple and effective bounding box representation using the idea of a polar coordinate system. It naturally circumvents the boundary discontinuity problem and generates regular boxes without post-processing. *Cheng et al. (2023)* proposed a spatial and channel converter to capture remote spatial interactions and critical correlations hidden in feature channels and designed a multi-interest region loss model based on a deep metric learning protocol to enhance fine-grained class separability.

The above algorithms enrich the solution ideas of remote sensing object detection tasks, but these algorithms often use deep convolutional networks to fit complex features, which makes the feature information of small objects easily lost when the feature information is extracted by the deep network. At the same time, direct fusion methods between multi-layer feature maps often also lead to more serious confusion on feature scales. To deal with these issues, an adaptive adjacent context negotiation network ($A^2$CN-Net) is given in this study.

## Visual attention mechanism

Attention mimics human attention to important information, focusing more on the essential aspects of the data by highlighting key details. Researchers have adopted similar concepts in many fields and have formulated various visual attention mechanisms to strengthen the performance of the algorithm. The main purpose of visual attention is to mimic the human visual cognitive system, which can be broadly categorized into Channel Attention (CA) and Spatial Attention (SA).

The CA realigns the importance of each channel to decide which information to focus on, just like when picking an item. This mechanism automatically selects the most important information for better data processing. *Hu, Shen & Sun (2018)* first introduced the concept of channel attention, which centers on a Squeeze-and-Excitation (SE) module. This module automatically adjusts the weighting parameters for each channel to better capture the most important information. Since the SE module can only use global average pooling to collect limited global information, this may lead to information loss and computational bottleneck problems. To solve these problems, *Dai et al. (2019)* used global 2D average pooling to improve the squeezing module by reflecting the relationship between channels in the form of covariance. *Wang et al. (2020b)* proposed an efficient channel attention, which replaces the global average pooling with a one-dimensional convolutional, and thus it can more efficiently capture global information and avoid information loss and computational bottleneck problems. Since the SE attention module only focuses on the channel dimension, it cannot capture the relationship between frequency channels well. For this reason, *Qin et al. (2021)* introduced the frequency attention module to adaptively adjust the weights of each frequency channel to better capture the most important information in each frequency channel.

The role of SA is to adjust the weights of each position of the image to focus the model more on those important regions. Generally, CA and SA are used in combination in tandem or parallel, and the most representative one is the Convolutional Block Attention Module (CBAM) (*Woo et al., 2018*). CBAM cascades CA and SA in two independent dimensions (channel and spatial) and obtains the attention map by global average pooling and maximum pooling. Then, the attention weights are multiplied by the input feature map for pixel weighting. The Bottleneck Attention Module (BAM) (*Park et al., 2018*) uses a parallel approach to integrate CA and SA. *Fu et al. (2019)* proposed a dual attention network for scene segmentation, which introduces the CA and SA with an adaptive gating mechanism, which allows the model to select and adjust the attention weights more flexibly, thus making the model more accurately process images of different scenes.

In the field of RSIs, *Li et al. (2023)* designed a supervised attention module to re-weight feature matrices at different scales respectively to efficiently aggregate multi-level features from high to low levels. *Liang et al. (2023)* designed a hybrid attention mechanism that combines the advantages of CA and SA to learn the spatial dependence of each channel and obtain richer critical high-frequency information. *Gu et al. (2023)* cleverly merged the non-local modules of self-attention into a unified unit, thus solving the problem of information loss when implementing SA and CA respectively. *Liu et al. (2023)* proposed to combine multiscale convolution with nonlocal spatial attention to construct more efficient

multiscale-based nonlocal perceptual fusion networks. The above methods introduce the attention mechanism in RSOD, which boosts the detection performance of the detector to some degree, and provides a solid theoretical and experimental basis for subsequent research, but limited by the characteristics of the RSIs, there is still a lot of work to be done on how to utilize the attention mechanism to extract more discriminative features to be explored.

# PROPOSED METHOD

The schematic diagram in Fig. 1 illustrates the framework of $A^2$CN-Net. The principle of fast Fourier transform is first used to design a Composite Fast Fourier Convolution module (CFFC), which is applied to each stage of the backbone network to perceive the spectrum context information and extract more abundant small target feature information. Secondly, based on self-attention theory, the global context information enhancement module is proposed to obtain rich spatial context information. Finally, to take full advantage of the feature correlation between adjacent feature layers, an adaptive adjacent context negotiation network is given to obtain more discriminative features. The network consists of local branches and adjacent branches. The local branches adaptively highlight the feature information, and the adjacent branches introduce global information into the adjacent layer to boost the feature representation and compensate for the information loss. Considering the different focuses of the feature layers at different dimensions, learnable weights are applied to the local and adjacent branches for feature fusion.

## Composite fast fourier convolution

The Fourier Transform (FT) has been widely utilized in the field of image processing due to its powerful analytical capabilities. FT is a mathematical tool that can decompose a signal or image into a series of frequency components, thus providing a detailed description of the signal or image frequency domain features. Inspired by this, recently a neural operator Fast Fourier Convolution (FFC) (_Chi, Jiang & Mu, 2020_), has been proposed. The introduction of FFC provides a powerful tool for neural networks to utilize the global contextual information in an image more efficiently, improving the performance and accuracy of image processing tasks. By using FFC, neural networks can analyze the entire image at a global level, capturing a wider range of contextual information. Compared to traditional convolution, FFC enables a larger range of sensory fields, leading to a better understanding of the global structure and associations in the image. This ability for non-local inference and generation allows neural networks to better handle long-range dependencies and global features in images. In this article, we introduce FFC into the RSOD task and further enhance the feature information of the target through frequency domain convolution to facilitate sensitivity to small objects. The idea of the CSPNet (_Wang et al., 2020a_) structure is used to design a Composite Fast Fourier Convolution (CFFC) module.

As presented in Fig. 2, for the CFFC module, the input feature layer is divided into two parts. The upper branch uses CondConv, BN layer, and ReLU layer to extract local deep refinement feature information. The lower branch uses dynamic convolution to obtain feature information for large-scale objects and reduce the resolution of the feature

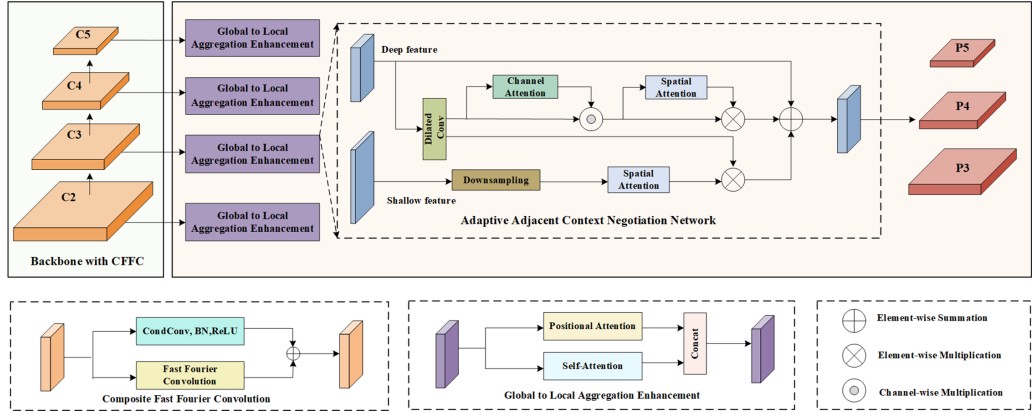

**Figure 1** **Network structure of $A^2$CN-Net.** It first applies the backbone (modified by CFFC) to obtain multi-scale features C2, C3, C4, C5. Then, the Global Context Information Enhancement (GCIE) module is given to capture and aggregate global spatial features, which is beneficial for locating targets of different sizes. After that, the Adaptive Adjacent Context Negotiation network is used to obtain multi-scale features P3, P4, P5 for detection.

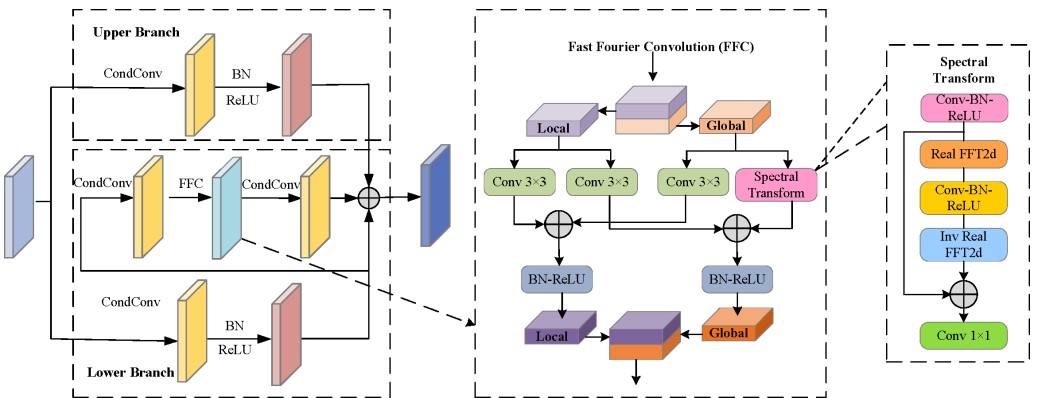

**Figure 2** **Illustration of Composite Fast Fourier Convolution (CFFC).**

map, and then FFC is introduced to extract the feature information. Finally, the two branches are merged using the cross-stage hierarchy structure, which enables the network to achieve a richer gradient combination. For the FFC module, the local branch of FFC uses traditional convolution for feature updating, and the global branch of FFC uses Fourier transform for spectral updating. Specifically, first, FFC applies Real FFT2d to the frequency dimension of the input feature graph and concatenates the real and imaginary parts of the spectrum in the channel dimension. Second, convolution blocks (including convolution, normalization, and activation functions) are applied in the frequency domain. Finally, the inverse transform is applied to recover the spatial structure.

## Global context information enhancement

Remote sensing satellites are higher from the ground and have a wide field of view. Therefore, RSIs contain larger scenes and more abundant information. Usually, these messages have strong semantic information, which can represent the scene of the target and form the prior knowledge of the object. In a scene, semantic information is usually inseparable from the objects in the scene. Objects in a scene carry a wealth of semantic information in their location, shape, size, color, and so on. Together, these objects constitute the visual content of the scene and convey semantic meanings through their features and interrelationships. By observing and understanding the objects in a scene, we can infer the nature of the scene, the environment, the activity, or the context. Thus, for scene comprehension and semantic reasoning, objects play a crucial role in a scene; they are important cues for decoding and interpreting the semantics of a scene. For example, the ship generally stays on the sea or the port, the aircraft stays at the airport, the windmill is mostly located in the desolate Gobi Beach, and the vehicles in the parking lot are orderly and densely distributed. Typically, small objects occupy relatively limited pixels and space in an image or scene, and their representation at the pixel level usually involves only a small amount of pixel points, so it is necessary to capture the context information around the small object or further away to highlight the location information of the small object. By the above findings, we combined the basic knowledge of self-attention and location-attention mechanisms to design a global context information enhancement module.

As shown in Fig. 3, the global context information enhancement module consists of the upper and lower branches splicing on the channel. The upper branch mainly learns from the self-attention mechanism, which first divides the input features into different image patches, and then uses the multi-head attention mechanism (*Vaswani et al., 2017*) to realize the feature interaction between patches, to obtain the global long-term dependency. Specifically, the upper branch consists of layer normalization (LN), multi-head self-attention (MSA), residual structure, and multi-layer perception (MLP). The lower branch captures distant spatial dependencies by directly calculating the correlation between two locations. Specifically, first of all, the input feature map is linearly mapped to produce $\theta$, $\phi$, and g eature map, then the similarity of $\theta$, and $\phi$ is calculated, and the autocorrelation feature is normalized to obtain the position relation weight. At the same time, the feature matrix is processed by parallel dilated convolution of different convolution kernels to expand the perceptual field of view. Finally, it is multiplied by the position attention weight to get the output feature.

## Adaptive adjacent context negotiation

With the deepening of the CNN, for small objects, since they occupy fewer pixels, the amount of information they carry is relatively small. In the process of feature extraction, such relatively small objects often face the risk of feature loss. This is because the convolutional neural network may blur or weaken the subtle features of small objects when performing convolution and pooling operations, resulting in their information not being adequately captured and represented. Therefore, multi-scale features are often used for object detection in RSIs, typically, shallow features are mainly utilized for sensing

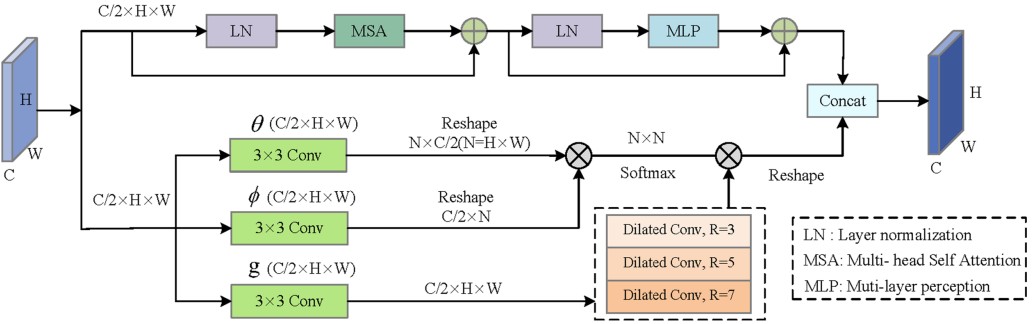

**Figure 3** **Illustration of Global Context Information Enhancement (GCIE).**

small objects, while deep features are used for sensing large objects. Researchers often employ various feature fusion strategies to mitigate information loss, such as the top-down feature pyramid, this can introduce redundant interfering information in the feature map, leading to feature confusion between multi-scale objects, thus reducing the effectiveness in detecting small targets.

To deal with the outlined issues, we design an adaptive adjacent upper-lower-layer coordination network to capture more discriminative features. As illustrated in Fig. 4, $A^2$CN is made up of local branches and adjacent branches. Local branches adaptively highlight feature information, and adjacent branches introduce global information into the feature layer to boost feature representations. The whole process can be expressed as:

$$F_o = \alpha F_2 + \beta F_3 + \chi F_i \tag{1}$$

where $F_{i-1}$, $F_i$ and $F_o$ represent shallow feature, deep feature and output feature respectively. $F_2$ and $F_3$ represent the feature of deep feature after local branch processing and the feature of deep feature after adjacent branch processing, respectively. $\alpha$, $\beta$, and $\chi$ represent learnable weight matrices.

Specifically, for local branches, multi-scale dilated convolution is first utilized to extract the input features using the multi-scale cavity convolution, and then the features are obtained by splicing the channel dimensions. Finally, It will be sent to channel attention and spatial attention to assign feature map weights, extracting key information while suppressing irrelevant information. The operation can be described as:

$$F_1 = Concat\left(DConv(F_i; W_{3\times3}^l, r^l)\right), 1 \in \{3, 5, 7\} \tag{2}$$

$$F_3 = SA(F_1 \odot CA(F_1)) \otimes (F_1 \odot CA(F_1)) \tag{3}$$

$$CA = \sigma\left(W_1 \times \frac{1}{H \times W} \sum_{m=1}^{H} \sum_{n=1}^{W} F_{1m,n}\right) \tag{4}$$

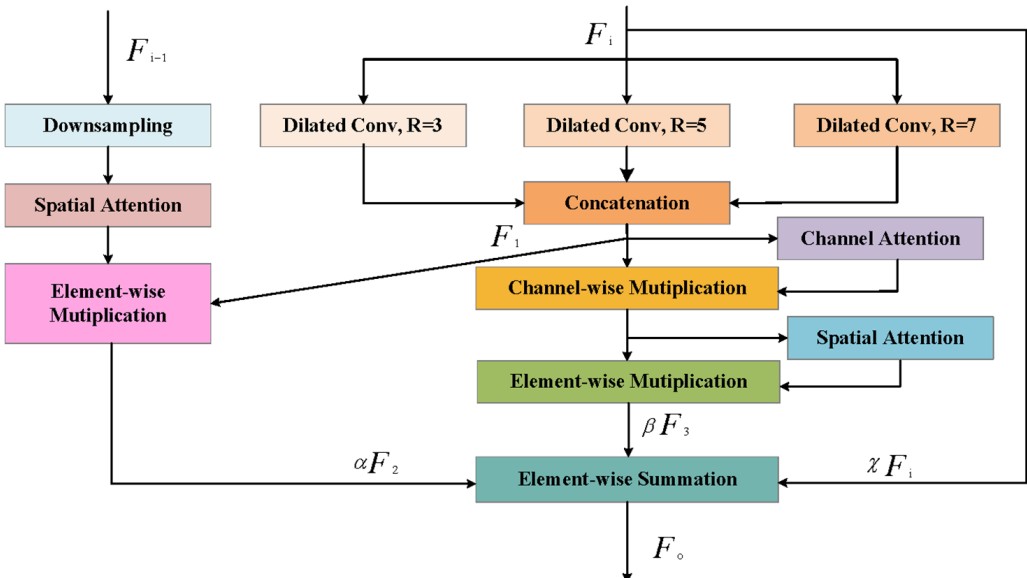

**Figure 4** Illustration of Adaptive Adjacent Context Negotiation ($A^2$CN).

$$SA = Conv\left(GMP\left(F_1 \odot CA\left(F_1\right)\right)\right) \tag{5}$$

where $Concat\left(.\right)$ is the cross-channel concatenation, $DConv\left(.\right)$ is the dilated convolution, $W^n_{3\times 3}$ is the parameters with $3\times 3$ kernel, $r$ is the dilation rate. In addition, $CA$ and $SA$ represents channel attention and spatial attention, $\odot$ is the channel wise multiplication, and $\otimes$ is the elementwise multiplication. $H$ and $W$ are the height and width of the input feature map $F_1$ respectively; $W_1$ is the weight matrix, and $\sigma\left(.\right)$ is the sigmoid function. $GMP\left(.\right)$ is global max pooling.

For adjacent branches, the feature map resolution is adjusted first by using upper use, and then each pixel is weighted by using spatial attention. The process can be expressed as:

$$F_2 = F_1 \otimes SA\left(Down\left(F_{i-1}\right)\right) \tag{6}$$

where $Down\left(.\right)$ is the $2\times$ down sampling implemented by max-pooling.

To take full advantage of the feature correlation between adjacent feature layers, the information loss caused by the target after sampling is compensated. Adaptive adjacent upper and lower coordination network fuses feature of local fraction and adjacent branches. Meanwhile, for the feature confusion resulting from the direct superposition of feature layers with different depths, we adopt a learnable weight coefficient to alleviate the influence.

# EXPERIMENTS

## Datasets

The DIOR dataset (*Li et al., 2020*) and DOTA dataset (*Xia et al., 2018*) are used to validate the effectiveness of our algorithms, both of which are large-scale datasets widely used for

object detection in aerial imagery. The DIOR dataset is not only large in image size, but also contains a wide range of different imaging conditions, including weather, seasons, and image qualities, a wide range of inter- and intra-class size variations of the objects, and a high degree of inter-class similarity and intra-class diversity, which gives the dataset a rich variety of image diversity and variability. The DOTA dataset comes from a variety of sources, including aerial imagery collected by different sensors and platforms, which makes the dataset rich in scenario variations and practical applications. Secondly, the targets in the images have a wide range of scale, orientation, and shape variations, which provides a great challenge for object detection algorithms. The use of two different datasets enables the performance of the model to be evaluated in different domain conditions and object classes, providing a more comprehensive understanding of its effectiveness. Both datasets are described in detail next.

1. DIOR: The DIOR dataset has a total of 23,463 remote sensing images and 190,288 object instances, comprising 20 object classes, as shown in Table 2, each object category in DIOR is assigned an index (CR1-CR20). We selected 5,862 images as the training set, 5,863 images as the verification set, and 11,738 images as the test set.

2. DOTA-v1.0: The DOTA-v1.0 dataset contains 2,806 images, each with a pixel size ranging from 800 × 800 to 4,000 × 4,000, in a total of 15 categories, as shown in Table 3, each object category is assigned an index (CA1-CA20). Since the DOTA-V1.0 dataset contains images with large resolution, the distribution of objects in the images is uneven and there are a lot of background regions, so in this study, standard development tools of DOTA are uniformly adopted to cut the images, and the cutting step is set to 256. After cutting, the output image size is 1,024 × 1,024, and data cleaning is performed. The cut data set comprises a comprehensive collection of 25,391 images, 15,235 images designated for the training set, 5,078 images allocated for the verification set, and an additional 5,078 images specifically reserved for the test set. In this article, objects whose width and height are both less than 50 pixels are considered small objects (*Wang et al., 2019*). Concerning this standard, Tables 2 and 3 analyze the proportion of small objects for each category in the DIOR and DOTA-v1.0 datasets. It is not difficult to find from the table that both of these two public data sets contain a large amount of remote sensing small target examples, and the experiment results of small targets are persuasive to a certain extent when conducted on this dataset.

## Implementation details and evaluation metrics

In this article, the basic learning rate is set to 0.01, the training epochs are set to 150, the batch size is set to 32, and the size of each image is adjusted to 800 × 800. A stochastic gradient descent algorithm (SGD) is adopted to optimize the parameters of the model. The initial learning rate of SGD is set to 0.01. The input images are preprocessed by random rotation, Mosaic data enhancement, random cropping, and color dithering. During the test, the NMS is used to detect the enclosure. The confidence threshold is set to 0.5, and the threshold of the bounding enclosure IoU is set to 0.5 when the AP is calculated. When testing the detection results of the model, the NMS confidence threshold is set to 0.5. The selection of these parameters was based on a comprehensive analysis of the

**Table 2  Detailed information on each class of the DIOR dataset.**

|  | CR1 | CR2 | CR3 | CR4 | CR5 |
|---|---|---|---|---|---|
| Class | Airplane | Airport | Baseball field | Basketball court | Bridge |
| Small object | 5712 | 1 | 1185 | 202 | 2070 |
| Total object | 10104 | 1327 | 5815 | 3225 | 3965 |
| Ratio | 0.565 | 0.0007 | 0.203 | 0.062 | 0.522 |
|  | CR6 | CR7 | CR8 | CR9 | CR10 |
| Class | Chimney | Dam | Expressway service area | Expressway toll station | Golf field |
| Small object | 105 | 74 | 145 | 498 | 0 |
| Total object | 1681 | 1049 | 2165 | 1298 | 1086 |
| Ratio | 0.062 | 0.0705 | 0.066 | 0.383 | 0 |
|  | CR11 | CR12 | CR13 | CR14 | CR15 |
| Class | Ground track field | Harbor | Ship | Stadium | Storage tank |
| Small object | 806 | 721 | 55201 | 74 | 21080 |
| Total object | 3038 | 5455 | 62157 | 1268 | 26262 |
| Ratio | 0.265 | 0.132 | 0.888 | 0.058 | 0.802 |
|  | CR16 | CR17 | CR18 | CR19 | CR20 |
| Class | Tennis court | Train station | Vehicle | Windmill | Overpass |
| Small object | 2623 | 2 | 36035 | 3493 | 884 |
| Total object | 12260 | 1011 | 40304 | 5363 | 3112 |
| Ratio | 0.213 | 0.001 | 0.894 | 0.651 | 0.284 |

**Table 3  Detailed information on each class of the DOTA-v1.0 dataset.**

|  | CA1 | CA2 | CA3 | CA4 | CA5 |
|---|---|---|---|---|---|
| Class | Plane | Baseball diamond | Bridge | Ground track field | Small vehicle |
| Small object | 13223 | 478 | 6546 | 343 | 93217 |
| Total object | 47187 | 2266 | 8953 | 2342 | 94408 |
| Ratio | 0.28 | 0.21 | 0.731 | 0.146 | 0.987 |
|  | CA6 | CA7 | CA8 | CA9 | CA10 |
| Class | Large vehicle | Ship | Tennis court | Basketball court | Storage tank |
| Small object | 33414 | 117741 | 922 | 547 | 26792 |
| Total object | 67084 | 145420 | 8625 | 2387 | 31598 |
| Ratio | 0.498 | 0.809 | 0.106 | 0.229 | 0.847 |
|  | CA11 | CA12 | CA13 | CA14 | CA15 |
| Class | Soccer ball field | Roundabout | Harbor | Swimming pool | Helicopter |
| Small object | 414 | 1291 | 9333 | 5932 | 1967 |
| Total object | 2426 | 2122 | 31966 | 7628 | 2922 |
| Ratio | 0.17 | 0.608 | 0.291 | 0.777 | 0.673 |

literature, previous experimental results, and fine-tuning processes. Multiple experiments were conducted to optimize these hyperparameters and determine the most effective configurations for our object detection task. The hardware and software environments are shown in Table 4.

**Table 4  Hardware and software configuration.**

| Hardware | | | Software | | |
|---|---|---|---|---|---|
| CPU | Intel Xeon E5-2698 v4 | System | Ubuntu 18.04 | CUDA | 10.0 |
| GPU | Tesla V100 | Python | Python 3.7 | cudnn | 7.6.5 |
| Memory | 6T | Pytorch | Pytorch1.7.1 | | |

The experiments use mAP (Mean Average Precision) and AP (Average Precision) for each type of detection as evaluation metrics, which is the main evaluation metric in the field of object detection. AP represents the average precision under different recall rates. Precision (P) refers to the ratio of the number of correctly detected samples to the total number of detections, and Recall (R) refers to the ratio of the number of correctly detected samples to the number of all true value samples, and mAP is the average value of all categories of APs. The formulas are as follows:

$$P = \frac{TP}{TP + FP} \tag{7}$$

$$R = \frac{TP}{TP + FN} \tag{8}$$

$$AP = \int_0^1 P(R)dR \tag{9}$$

$$mAP = \sum_{i=1}^{N} Ap_i / N \tag{10}$$

where TP represents the number of correctly classified samples in the class, FP represents the number of samples from other classes that were incorrectly identified as samples from the class, and FN represents the number of samples from the class that were incorrectly identified as samples from other classes.

## Results for DIOR dataset and DOTA-v1.0 dataset

Tables 5 and 6 present the performance metrics of our $A^2$CN-Net architecture on the DIOR and DOTA-v1.0 datasets, respectively. These tables highlight the average accuracy (AP) values for each category within the datasets, providing a quantitative assessment of the model's ability to accurately classify objects across various classes. Additionally, the tables include precision and recall values for each category on both datasets. As shown in Table 5, there are 11 categories of targets with AP values above 75%, and only CR5 (bridge) has an AP value below 60%. By looking at the confusion matrix, almost more than half of the targets like bridges are missed. As shown in Table 3, more than half of the targets with small sizes among the targets like bridges bring a great challenge to the detection task. As shown in Table 6, there are 11 categories of targets with AP values over 75%, and CA6

**Table 5 AP values for the DIOR dataset.**

| Class | CR1 | CR2 | CR3 | CR4 | CR5 | CR6 | CR7 | CR8 | CR9 | CR10 | |
|---|---|---|---|---|---|---|---|---|---|---|---|
| Precision (%) | 97.2 | 93.1 | 95.6 | 92.8 | 67.1 | 94.4 | 74.3 | 94.8 | 94.4 | 87.9 | |
| Recall (%) | 70.8 | 63.0 | 63.1 | 73.1 | 36.3 | 67.7 | 50.7 | 39.0 | 49.0 | 68.5 | |
| AP (%) | 84.6 | 79.5 | 80.5 | 85.0 | 53.2 | 82.8 | 62.4 | 67.7 | 72.7 | 79.3 | |
| Class | CR11 | CR12 | CR13 | CR14 | CR15 | CR16 | CR17 | CR18 | CR19 | CR20 | All |
| Precision (%) | 76.7 | 77.6 | 94.8 | 90.7 | 93.7 | 97.0 | 72.1 | 79.1 | 86.0 | 74.5 | 86.7 |
| Recall (%) | 66.4 | 53.5 | 78.9 | 30.7 | 64.1 | 76.2 | 56.1 | 43.2 | 76.7 | 46.8 | 58.7 |
| AP (%) | 74.5 | 67.6 | 87.9 | 61.3 | 79.9 | 87.3 | 65.9 | 63.2 | 84.8 | 63.9 | 74.2 |

**Table 6 AP values for the DOTA-v1.0 dataset.**

| Class | CA1 | CA2 | CA3 | CA4 | CA5 | CA6 | CA7 | CA8 |
|---|---|---|---|---|---|---|---|---|
| Precision (%) | 95.1 | 81.1 | 73.8 | 86.2 | 72.8 | 82.1 | 91.8 | 95.5 |
| Recall (%) | 63.6 | 80.4 | 63.8 | 70.6 | 61.8 | 47.5 | 59.1 | 69.9 |
| AP (%) | 80.6 | 86.6 | 71.1 | 79.6 | 70.5 | 66.7 | 76.9 | 84 |
| Class | CA9 | CA10 | CA11 | CA12 | CA13 | CA14 | CA15 | All |
| Precision (%) | 91.6 | 92.2 | 87.2 | 85.3 | 90.5 | 85.6 | 77.5 | 85.9 |
| Recall (%) | 63.4 | 82.1 | 64.2 | 79.4 | 63.7 | 86.1 | 55.6 | 67.4 |
| AP (%) | 79.4 | 89.6 | 76.8 | 85.8 | 78.7 | 89.5 | 71.7 | 79.2 |

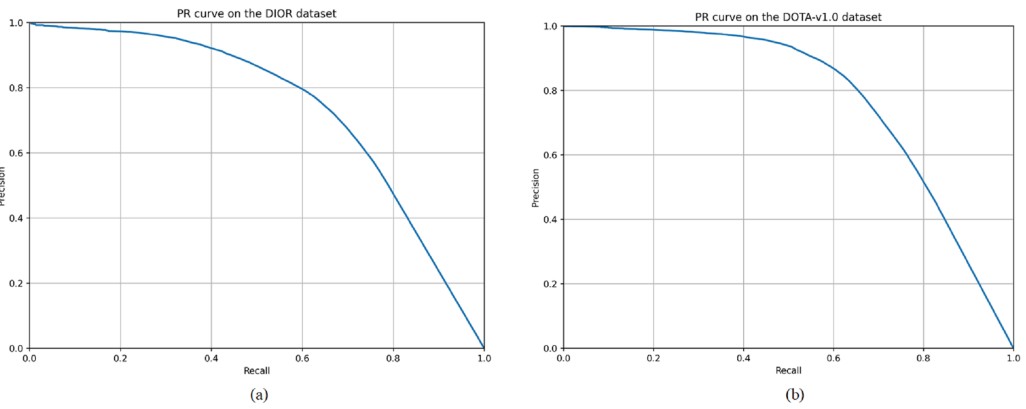

**Figure 5 Precision–recall curves of the $A^2$CN-Net for different classes on the DIOR (A) and DOTA-v1.0 (B) datasets, respectively.**

(large vehicle) has the lowest AP value of 66.7%. By looking at the confusion matrix, half of the targets among the large vehicles are missed, which is the worst missed detection among all the categories. Also, 3% of the large vehicles were mistakenly detected as small vehicles. Figure 5 shows the precision–recall curves of the proposed algorithm for different categories on the DIOR and DOTA-v1.0 datasets, respectively.

**Table 7  Comparison of ablation results in the DIOR dataset.**

| | CR1 | CR2 | CR3 | CR4 | CR5 | CR6 | CR7 | CR8 | CR9 | CR10 | |
|---|---|---|---|---|---|---|---|---|---|---|---|
| Baseline | 81 | 59.9 | 81.8 | 84.4 | 47 | 81.3 | 42.1 | 59.5 | 74.3 | 54 | |
| CFFC | 83.5 | 62.9 | 81.6 | 83.5 | 49.7 | 80.9 | 45.6 | 62.6 | 74.3 | 63.5 | |
| GCIE | 86.1 | 72 | 80.2 | 84.2 | 52.1 | 81.5 | 50.7 | 65.3 | 73.6 | 72.8 | |
| $A^2$CN | 86.4 | 70.9 | 81.6 | 84.7 | 49 | 81.8 | 48.7 | 66.1 | 72.5 | 72.1 | |
| CFFC+GCIE | 86.4 | 75.7 | 80.4 | 84.7 | 53.1 | 82.7 | 56.8 | 66.8 | 74.3 | 76.5 | |
| CFFC+ $A^2$CN | 83.7 | 74.2 | 80.1 | 84.9 | 51.5 | 81.8 | 54.9 | 66.8 | 71.5 | 76.6 | |
| GCIE+ $A^2$CN | 86.4 | 77.4 | 80.7 | 84.8 | 50.1 | 82.5 | 58.5 | 67.2 | 71.9 | 77.5 | |
| $A^2$CN-Net | 84.6 | 79.5 | 80.5 | 85 | 53.2 | 82.8 | 62.4 | 67.7 | 72.7 | 79.3 | |
| | CR11 | CR12 | CR13 | CR14 | CR15 | CR16 | CR17 | CR18 | CR19 | CR20 | mAP (%) |
| Baseline | 63.5 | 57.3 | 87.5 | 62.8 | 81.4 | 87.8 | 47.3 | 62.5 | 85.1 | 58.3 | 68.4 |
| CFFC | 65 | 59.8 | 87.7 | 70.6 | 81 | 87.3 | 44.7 | 60.8 | 85.6 | 60.8 | 69.6 |
| GCIE | 70.6 | 64.1 | 87.8 | 62.9 | 83 | 87.2 | 59.2 | 64 | 84.9 | 60.7 | 72.1 |
| $A^2$CN | 70 | 62.7 | 87.7 | 67.4 | 80.2 | 87.6 | 51.9 | 58.6 | 85.5 | 62.3 | 71.4 |
| CFFC+GCIE | 73.4 | 66.5 | 87.4 | 61.8 | 80.9 | 87.1 | 60.1 | 63 | 85.3 | 64.2 | 73.3 |
| CFFC+ $A^2$CN | 73 | 65.1 | 87.3 | 63 | 78.9 | 87 | 61.5 | 61.6 | 84.1 | 62.1 | 72.5 |
| GCIE+ $A^2$CN | 72 | 65 | 87.6 | 63.9 | 80.7 | 87.7 | 59.6 | 59.1 | 84.6 | 61.7 | 72.9 |
| $A^2$CN-Net | 74.5 | 67.6 | 87.9 | 61.3 | 79.9 | 87.3 | 65.9 | 63.2 | 84.8 | 63.9 | 74.2 |

## Ablation studies

Under the same experimental conditions, ablation experiments are performed to verify the effectiveness of the designed modules one by one. Table 7 shows the results of $A^2$CN-Net's ablation experiments in various categories on the DIOR dataset.

1. **Baseline setup:** The baseline network is a single-stage detector, including a backbone network, feature fusion, and multi-scale detection head. The baseline network is a single-stage detector, including a backbone network feature fusion, and multi-scale detection head. The backbone network uses the general Resnet50 network, due to the ResNet50 model has been widely adopted for many computer vision tasks including remote sensing object detection. It offers a powerful and effective feature extraction capability, enabling the model to learn high-level representations from input images. Its depth and complexity make it particularly suitable for capturing intricate patterns and context in the visual domain. The feature fusion part only uses the FPN structure. For the detection layer, CIOU is used as boundary box regression loss, and binary cross-entropy is used for classification loss. From the results of the baseline experiment in Table 7, it is not difficult to find that using a simple network structure while minimizing the use of up and downsampling can effectively alleviate the information loss of small objects, such as the detection accuracy of 81% for airplanes (CR1), 87.5% for ships (CR13), and 85.1% for windmills (CR19). However, for objects with more complex backgrounds and large-scale changes, the detection accuracy can be improved, such as airports (CR2), bridges (CR5), dams (CR7), golf fields (CR10), ground track field (CR11), and overpasses (CR20).

2. **Compound fast Fourier convolution (CFFC):** On the basis of the fast Fourier transform principle, the CFFC module fuses the frequency domain convolution and spatial convolution to make full use of the spatial structure information and spectrum context information of the target, effectively enhancing the feature representation of the small object. Compared with the results of the benchmark experiment, the insertion of FFC modules in different stages of the backbone network effectively improves the detection effect of small targets, such as the detection accuracy of airplanes (CR1) from 81% to 83.5%, the detection accuracy of bridges (CR5) from 47% to 49.7%, and the detection accuracy of ships (CR13) and windmills (CR19) has been slightly improved. The CFFC module improves the overall average accuracy by 1.2% mAP.

3. **Global context information enhancement module (GCIE):** For remote sensing images, small targets are often densely distributed together, resulting in blurred edges between objects and easy-to-miss detection. The GCIE module captures global background information by mapping the degree of association between pixels. Simultaneously, the receptive field is expanded by dilated convolution with different dilatation rates. This not only reduces the problem of missing small objects. For example, the detection accuracy of airplanes (CR1), storage tanks (CR15), bridges (CR5), and vehicles (CR18) have been improved by 5.1%, 1.6%, 4.1%, and 1.5%, respectively. It is also very helpful for objects with large-scale changes and objects with complex backgrounds. For example, the detection accuracy of airports (CR2), expressway service areas (CR8), golf fields (CR10), ground track fields (CR11), and train stations (CR17) are increased by 12.1%, 5.8%, 8.8%, 7.1%, and 11.9%, respectively. The GCIE module improves the overall average accuracy by 3.7% mAP.

4. **Adaptive adjacent context negotiation network ($A^2$CN):** Since shallow feature and deep feature have their advantages and shortcomings in the detection of small and large targets, how to comprehensively utilize the characteristics of the two becomes the key to optimizing performance and robustness of target detection. The $A^2$CN module fully uses the feature correlation between adjacent layers and uses an attention mechanism to suppress context-independent information and highlight the relevant information of target features. Simultaneously, considering the different focus of feature maps of different scales, we do not simply add feature pixels of different scales directly, but assign learnable weight matrices to different branches for adaptive feature fusion. By observing Table 7, the $A^2$CN module enhances the detection capability of most categories, among which the improvement effect is obvious for airplanes (CR1), airports (CR2), golf fields (CR10), ground track fields (CR11), and harbors (CR12). The $A^2$CN module improves the overall average accuracy by 3% mAP.

5. **The algorithm proposed in this study ($A^2$CN-Net):** Based on the baseline network, CFFC, GCIE, and $A^2$CN modules are used simultaneously. It can be observed from Table 7 that aircraft, Bridges, and ships with a relatively high proportion of small objects have all improved, increasing by 5.4%, 2%, and 0.4% respectively. airports (CR2), expressway service areas (CR8), golf fields (CR10), harbors (CR12), ground track fields (CR11), train stations (CR17), and dams (CR7) with high objects have been significantly improved, increasing by 20%, 6.6%, 8.1%, 5.4%, 6.5%, 4.6%, and 6.6%

respectively. The $A^2$CN-Net improves the overall average accuracy by 5.8% mAP. In summary, $A^2$CN-Net has a good detection effect on multi-class remote sensing objects of different scales.

## Comparisons with other approaches

$A^2$CN-Net is used to compare results with mainstream detectors on the DIOR dataset. Among them, the detectors involved in the comparison include Two-stage methods: Faster RCNN (*Ren et al., 2015*), CSFF (*Cheng et al., 2020*), GLNet (*Teng et al., 2021*), and MFPNet (*Yuan et al., 2021*). One-stage methods: SRAFNet (*Liu et al., 2021a*), AFDet (*Liu et al., 2021b*), FFPFNet (*Lingyun, Popov & Ge, 2022*), GAB-Net (*Zhang et al., 2023*), QETR (*Ma, Lv & Zhong, 2024*), and SRARNet (*Yang & Wang, 2024*).

Table 8 gives the detection accuracy of different algorithms for different objects on the DIOR dataset. It is not difficult to find that the Faster RCNN has the worst detection ability. This is mainly due to Faster RCNN's low detection performance for small target objects (*e.g.*, windmill (CR19) AP value is only 5.3%) and poor ability to deal with intra-class similarity problems (*e.g.*, bridges (CR5) and overpasses (CR20)). The mAP values of CSFF, GLNet, MFPNet, and GAB-Net are all higher than 65%, which is mainly because they adopt multi-scale feature fusion or multi-scale feature prediction, which makes full use of the feature information and thus significantly improves the accuracy of the detectors. SRAFNet shows excellent detection results compared to other networks in multiple categories such as airplanes (CR1), baseball fields (CR3), chimneys (CR6), and windmills (CR19). However, the detection effect of this detector on some targets is poor. For example, ground track fields (CR11) and bridges (CR5) have AP of only 16.2% and 35.8%, respectively. In the comparison, the algorithms presented in this article perform better, with AP values of 74.5% and 53.2%, respectively. QETR uses multiple self-attention and cross-attention mechanisms to learn object features and location information, which significantly improves detection accuracy for targets with large scales, *e.g.*, airport (CR2), harbor (CR12), stadium (CR14), and train stations (CR17). However, QETR has lower detection accuracy for smaller targets. For example, storage tank (CR15) and vehicle (CR18). And our algorithm has more than 60% detection accuracy in these two categories. AFDet uses an elliptic Gauss kernel to generate key point heat maps to adapt to targets with large aspect ratios and shows good detection performance for tennis courts (CR16), vehicles (CR18), and windmills (CR19). FFPFNet uses a bilateral spectrum-aware feature pyramid network to enhance feature extraction from objects, which achieves the best detection performance for the dam (CR7), expressway service area (CR8), and golf field (CR10). SRARNet combines super-resolution techniques with dynamic feature fusion to improve detection accuracy, using a super-resolution network with soft thresholding to refine small target features and shows the best detection results on several categories, including basketball court (CR4), expressway toll station (CR9), ship (CR13), storage tank (CR15), and tennis court (CR11).

Although these detection algorithms effectively enhance the precision of remote sensing objects through multi-scale fusion, multi-scale detection, thermal map-guided attention, and other methods, the effectiveness of these algorithms in detecting multi-scale and

**Table 8 AP for different algorithms on the DIOR dataset.**

| Method | CR1 | CR2 | CR3 | CR4 | CR5 | CR6 | CR7 | CR8 | CR9 | CR10 | |
|--------|-----|-----|-----|-----|-----|-----|-----|-----|-----|------|--|
| | | | | | Two-stage methods | | | | | | |
| Faster RCNN | 37.2 | 62.2 | 64.1 | 70.9 | 20.6 | 72.4 | 45.8 | 56.5 | 42.9 | 69.6 | |
| CSFF | 57.2 | 79.6 | 70.1 | 87.4 | 46.1 | 76.6 | 62.7 | 82.6 | 73.2 | 81.6 | |
| GLNet | 62.9 | 83.2 | 72.0 | 81.1 | 50.5 | 79.3 | 67.4 | 86.2 | 70.9 | 83.0 | |
| MFPNet | 76.6 | 83.4 | 80.6 | 82.1 | 44.3 | 75.6 | 68.5 | 85.9 | 63.9 | 77.2 | |
| | | | | | One-stage methods | | | | | | |
| SRAFNet | 88.4 | 76.5 | 92.6 | 87.9 | 35.8 | 83.8 | 58.6 | 86.8 | 66.8 | 82.8 | |
| QETR | 73.3 | 90.3 | 77.6 | 88.2 | 47.2 | 82.5 | 76.6 | 86.9 | 72.8 | 84.5 | |
| AFDet | 82.4 | 81.5 | 81.9 | 89.8 | 51.7 | 74.9 | 58.7 | 84.2 | 73.3 | 81.0 | |
| FFPFNet | 65.5 | 86.7 | 79.4 | 89.0 | 50.3 | 79.2 | 73.3 | 87.6 | 73.6 | 85.1 | |
| GAB-Net | 85.9 | 83.3 | 84.4 | 89.1 | 44.9 | 78.7 | 67.2 | 66.7 | 64.0 | 74.5 | |
| SRARNet | 88.1 | 69.6 | 84.6 | 91.6 | 50.0 | 79.4 | 49.5 | 78.2 | 77.3 | 80.5 | |
| $A^2$CN-Net | 84.6 | 79.5 | 80.5 | 85.0 | 53.2 | 82.8 | 62.4 | 67.7 | 72.7 | 79.3 | |
| Method | CR11 | CR12 | CR13 | CR14 | CR15 | CR16 | CR17 | CR18 | CR19 | CR20 | mAP (%) |
| | | | | | Two-stage methods | | | | | | |
| Faster RCNN | 41.6 | 42.6 | 46.3 | 7.0 | 64.1 | 13.8 | 60.9 | 45.5 | 5.3 | 30.8 | 45.1 |
| CSFF | 50.7 | 78.2 | 73.3 | 63.4 | 58.5 | 85.9 | 61.9 | 42.9 | 86.9 | 59.5 | 68.0 |
| GLNet | 51.8 | 81.8 | 72.0 | 75.3 | 53.7 | 81.3 | 65.5 | 43.4 | 89.2 | 62.6 | 70.7 |
| MFPNet | 62.1 | 77.3 | 77.2 | 76.8 | 60.3 | 86.4 | 64.5 | 41.5 | 80.2 | 58.8 | 71.2 |
| | | | | | One-stage methods | | | | | | |
| SRAFNet | 16.2 | 76.4 | 59.4 | 80.9 | 55.6 | 90.6 | 52.0 | 53.2 | 91.0 | 58.0 | 69.7 |
| QETR | 51.0 | 86.4 | 51.8 | 84.2 | 39.7 | 85.0 | 71.8 | 39.3 | 85.9 | 62.2 | 71.5 |
| AFDet | 44.2 | 79.5 | 77.8 | 63.2 | 76.9 | 91.0 | 62.0 | 59.3 | 87.1 | 62.0 | 73.2 |
| FFPFNet | 57.3 | 83.5 | 74.1 | 78.4 | 59.3 | 88.6 | 71.0 | 43.3 | 87.4 | 63.5 | 73.8 |
| GAB-Net | 64.7 | 78.2 | 91.1 | 74.1 | 78.4 | 90.9 | 62.1 | 54.7 | 81.8 | 60.3 | 73.8 |
| SRARNet | 59.7 | 68.2 | 91.8 | 78.2 | 81.1 | 93.1 | 45.4 | 60.2 | 90.9 | 60.0 | 73.9 |
| $A^2$CN-Net | 74.5 | 67.6 | 87.9 | 61.3 | 79.9 | 87.3 | 65.9 | 63.2 | 84.8 | 63.9 | 74.2 |

multi-category remote sensing objects could be improved. As shown in Table 8, $A^2$CN-Net has shown excellent detection results in several categories, such as bridges (CR5), ground track fields (CR11), vehicles (CR18), and overpasses (CR20). At the same time, $A^2$CN-Net has a more balanced detection effect across multiple categories, especially for objects with relatively few pixels, such as increasing the mAP of airplanes (CR1) to 84.6% and that of vehicles (CR18) to 63.2%. On the whole, the average detection accuracy of $A^2$CN-Net still maintains a high level.

To verify the generalization ability, $A^2$CN-Net is compared with the DOTA-v1.0 dataset containing more small objects. The detectors involved in the comparison include Two-stage methods: CAD-Net (*Zhang, Lu & Zhang, 2019*), SCRDet (*Yang et al., 2019*), FADet (*Li et al., 2019*), and APE (*Zhu, Du & Wu, 2020*). One-stage methods: APS-Net (*Zhou et al., 2022*), AFD (*Shamsolmoali et al., 2023*), CoF-Net (*Zhang, Lam & Wang, 2023*) and ASEM-Net (*Liu et al., 2024*). Table 9 gives the detection results of the different algorithms. To summarize, $A^2$CN-Net achieves the highest average detection accuracy and also shows

**Table 9   AP for different algorithms on the DOTA-v1.0 dataset.**

| Method | CA1 | CA2 | CA3 | CA4 | CA5 | CA6 | CA7 | CA8 |
|---|---|---|---|---|---|---|---|---|
| Two-stage methods | | | | | | | | |
| CAD-Net | 87.8 | 82.4 | 49.4 | 73.5 | 71.1 | 63.5 | 76.7 | 90.9 |
| SCRDet | 90.18 | 81.88 | 56.2 | 73.29 | 72.09 | 77.65 | 78.21 | 90.91 |
| FADet | 90.15 | 78.6 | 51.92 | 75.23 | 73.6 | 71.27 | 81.41 | 90.85 |
| APE | 89.96 | 83.62 | 53.42 | 76.03 | 74.01 | 77.16 | 79.45 | 90.83 |
| One-stage methods | | | | | | | | |
| ASEM-Net | 89.26 | 82.26 | 51.33 | 68.49 | 78.88 | 74.14 | 85.59 | 90.88 |
| AFD | 89.81 | 77.68 | 56.17 | 70.65 | 78.94 | 81.62 | 84.28 | 90.35 |
| CoF-Net | 89.6 | 83.1 | 48.3 | 73.6 | 78.2 | 83 | 86.7 | 90.2 |
| APS-Net | 89.75 | 81.26 | 58.12 | 72.84 | 80.74 | 83.29 | 88.05 | 90.9 |
| $A^2$CN-Net | 80.6 | 86.6 | 71.1 | 79.6 | 70.5 | 66.7 | 76.9 | 84 |
| Method | CA9 | CA10 | CA11 | CA12 | CA13 | CA14 | CA15 | mAP (%) |
| Two-stage methods | | | | | | | | |
| CAD-Net | 79.2 | 73.3 | 48.4 | 60.9 | 62 | 67 | 62.2 | 69.9 |
| SCRDet | 82.44 | 86.39 | 64.53 | 63.45 | 75.77 | 70.06 | 60.11 | 75.35 |
| FADet | 83.94 | 84.77 | 58.91 | 65.65 | 76.92 | 79.36 | 68.17 | 75.38 |
| APE | 87.15 | 84.51 | 67.72 | 60.33 | 74.61 | 71.84 | 65.55 | 75.75 |
| One-stage methods | | | | | | | | |
| ASEM-Net | 84.94 | 85.73 | 60.78 | 64.76 | 65.72 | 71.32 | 59.08 | 74.21 |
| AFD | 75.23 | 76.9 | 51.65 | 75.24 | 75.92 | 82.54 | 86.67 | 76.91 |
| CoF-Net | 82.3 | 86.6 | 67.6 | 64.6 | 74.7 | 71.3 | 78.4 | 77.2 |
| APS-Net | 86.19 | 86.41 | 65.26 | 67.39 | 76.65 | 74.49 | 65.44 | 77.79 |
| $A^2$CN-Net | 79.4 | 89.6 | 76.8 | 85.8 | 78.7 | 89.5 | 71.7 | 79.2 |

the best detection performance for multiple categories of targets, which mainly include: storage tanks (CA10), swimming pools(CA14), bridges (CA3), and helicopters (CA15). Carefully observing these categories, the detection accuracy of most of these categories improved by the $A^2$CN-Net is less than 80%, while the detection ability of other algorithms for these categories is worse, which directly indicates the effectiveness of the $A^2$CN-Net for detecting more difficult categories. In addition to enhancing the detection capability of small objects, the effect of large-size complex objects is also improved. $A^2$CN-Net shows good detection performance in all categories of the DOTA-v1.0 dataset, which fully proves its strong detection and classification ability.

# CONCLUSIONS

To tackle the problem of low detection accuracy of RSIs, this study analyzes the causes of small and medium-sized object information loss in convolutional neural networks in detail and proposes an $A^2$CN-Net detector based on adaptive adjacent context negotiation network. Based on the principle of fast Fourier transform, a composite fast Fourier convolution module is designed to extract the context information of the target on the spectrum and enhance the feature representation. Then, a global context information enhancement module based on a self-attention mechanism is designed to get rich spatial

context information. Secondly, considering the different focus of feature layers at different scales, an adaptive adjacent-context coordination network is designed to alleviate the feature aliasing problem caused by multi-scale feature fusion, and the features lost by small targets are replaced by the correlation of adjacent feature layers. Finally, the advances and effectiveness of $A^2$CN-Net are proved on DIOR and DOTA-v1.0 datasets. Although $A^2$CN-Net achieves good detection results in RSIs, it needs to sacrifice the parameter number and inference speed. Therefore, our next step is to look into efficient and lightweight solutions.

### Funding
This research was funded by NSFC (No. 62301623, No. 62072489), IRISTHN (211RISTHN013), Leading talents of Science and Technology in the Central Plain of China (234200510009), and Henan Province Key Science and Technology Research Projects (222102210008, 232102211002, 232102211030). The funders had no role in study design, data collection and analysis, decision to publish, or preparation of the manuscript.

### Grant Disclosures
The following grant information was disclosed by the authors:
NSFC: No. 62301623, No. 62072489.
IRISTHN: 211RISTHN013.
Leading talents of Science and Technology in the Central Plain of China: 234200510009.
Henan Province Key Science and Technology Research Projects: 222102210008, 232102211002, 232102211030.

### Competing Interests
The authors declare there are no competing interests.

### Author Contributions
- Yan Dong conceived and designed the experiments, performed the experiments, analyzed the data, performed the computation work, prepared figures and/or tables, authored or reviewed drafts of the article, and approved the final draft.
- Yundong Liu conceived and designed the experiments, performed the experiments, analyzed the data, performed the computation work, prepared figures and/or tables, authored or reviewed drafts of the article, and approved the final draft.
- Yuhua Cheng analyzed the data, prepared figures and/or tables, authored or reviewed drafts of the article, and approved the final draft.
- Guangshuai Gao conceived and designed the experiments, performed the experiments, analyzed the data, performed the computation work, prepared figures and/or tables, and approved the final draft.
- Kai Chen analyzed the data, authored or reviewed drafts of the article, and approved the final draft.

- Chunlei Li conceived and designed the experiments, performed the experiments, performed the computation work, prepared figures and/or tables, authored or reviewed drafts of the article, and approved the final draft.

## Data Availability

The code is available at GitHub and Zenodo:

- https://github.com/liuyundong-2020/A2CN-Net.

- liuyundong-2020. (2024). liuyundong-2020/A2CN-Net: v2. Zenodo. https://doi.org/10.5281/zenodo.11632568.

The DOTA dataset is an aerial image dataset produced by Gui-Song Xia of Wuhan University available at: https://captain-whu.github.io/DOTA/dataset.html.

The DIOR dataset is an aerial image dataset produced by Ke Li of Northwestern Polytechnical University available at: https://gcheng-nwpu.github.io/#Datasets.

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
