# Peer review of "Adaptive adjacent context negotiation network for object detection in remote sensing imagery"

_PeerJ Computer Science, doi:10.7717/peerj-cs.2199_

## Round 0.1 · original submission · Major Revisions

Dear authors,

You are advised to critically respond to all comments point by point when preparing a new version of the manuscript and while preparing for the rebuttal letter. Please address all the comments/suggestions provided by the reviewers.

Kind regards,
PCoelho

Reviewer 1 ·

Basic reporting

see attached file

Experimental design

see attached file

Validity of the findings

see attached file

Additional comments

see attached file

Annotated reviews are not available for download in order to protect the identity of reviewers who chose to remain anonymous.

Reviewer 2 ·

Basic reporting

All comments have been added in detail to the 4th section called additional comments.

Experimental design

All comments have been added in detail to the 4th section called additional comments.

Validity of the findings

All comments have been added in detail to the 4th section called additional comments.

Additional comments

Review Report for PeerJ Computer Science
(Adaptive adjacent context negotiation network for object detection in remote sensing imagery)

1. Within the scope of the study, a new deep learning-based model called Adaptive Adjacent Context Negotiation Network has been proposed to solve object detection problems in the field of remote sensing.

2. In the Introduction section, the importance of the problem addressed, its main contributions and its main originality points are mentioned.

3. Although the related works section is generally acceptable, adding a literature table consisting of sections such as dataset, model, results and metrics will bring the study to the fore.

4. In the Proposed method section, the Adaptive Adjacent Context Negotiation Network structure in Figure-1 is mentioned in detail and in a descriptive way. It is stated that the ResNet50 model is used as the backbone in the proposed network. Although there are many different backbones that can be used when the literature is examined, it should be explained more clearly why this particular one is preferred.

5. Composite Fast Fourier Convolution in Figure-2, Global Context Information Enhancement in Figure-3 and Adaptive Adjacent Context Negotiation in Figure-4 are understandable, acceptable and at a sufficient level in terms of originality.

6. In the study, 20-class DIOR and 15-class DOTA-v1.0, which are open source datasets frequently used in the field of remote sensing, were used as datasets. Applying the Adaptive Adjacent Context Negotiation Network developed within the scope of the study on more than one (two different) datasets instead of a single dataset is very important and valuable in terms of the applicability of the model and its comparison with the literature. However, experiments on different remote sensing datasets can further increase the value of the study. From this perspective, explain why these two datasets are preferred and/or why different datasets should not be used.

7. In the Implementation section, the hyperparameters used within the scope of the study are specified. The basis on which parameters such as learn rate, batch size, epoch, optimizer are selected or how they are selected greatly affects the object detection results. For this reason, it should be explained how these parameters were determined and whether different experiments were performed. Also, in the configuration section, it should be mentioned in more detail which toolbox/framework etc. is used.

8. It is recommended to interpret and compare the results obtained in the Comparisons section in terms of single-stage and two-stage detectors. In addition, although the models compared are suitable, it is recommended to add a few new detection models with more up-to-date versions.

9. In terms of evaluation metrics, missing metrics need to be added in order to analyze the results correctly. Such as precision-recall curve, Count of predicted bounding box, Average Recall (AR), Optimal Localization Recall Precision (oLRP).

10. In the detection results given in Figure 5, it is recommended to include separate results for each class for two different datasets.

As a result, although the study contains unique points in terms of the proposed model, the parts mentioned above should definitely be taken into consideration in order to improve the analysis of the results and other parts.

---

## Round 0.2 · accepted · Accept

Dear authors, we are pleased to verify that you meet the reviewer's valuable feedback to improve your research.

Thank you for considering PeerJ Computer Science and submitting your work.

Reviewer 1 ·

Basic reporting

The authors have well addressed my comments.

Experimental design

The authors have well addressed my comments.

Validity of the findings

The authors have well addressed my comments.

Additional comments

The authors have well addressed my comments.

Reviewer 2 ·

Basic reporting

All comments have been added in detail to the last section.

Experimental design

All comments have been added in detail to the last section.

Validity of the findings

All comments have been added in detail to the last section.

Additional comments

Review Report for PeerJ Computer Science
(Adaptive adjacent context negotiation network for object detection in remote sensing imagery)

Thanks for the revision. All revisions made to the paper and the final version of the paper have been examined in detail. Although some reviewer comments (some evaluation metrics) are not fully completed, I recommend that this research paper be accepted as it is due to both the originality of the study and its contribution to the literature. I wish the authors success in their future studies. Best regards.